# MisGAN: Learning from Incomplete Data with Generative Adversarial Networks

**Steven Cheng-Xian Li**
University of Massachusetts Amherst
cxl@cs.umass.edu

**Bo Jiang**
Shanghai Jiao Tong University
bjiang@sjtu.edu.cn

**Benjamin M. Marlin**
University of Massachusetts Amherst
marlin@cs.umass.edu

## Abstract

Generative adversarial networks (GANs) have been shown to provide an effective way to model complex distributions and have obtained impressive results on various challenging tasks. However, typical GANs require fully-observed data during training. In this paper, we present a GAN-based framework for learning from complex, high-dimensional incomplete data. The proposed framework learns a complete data generator along with a mask generator that models the missing data distribution. We further demonstrate how to impute missing data by equipping our framework with an adversarially trained imputer. We evaluate the proposed framework using a series of experiments with several types of missing data processes under the missing completely at random assumption.[1]

## 1 Introduction

Generative adversarial networks (GANs) (Goodfellow et al., 2014) provide a powerful modeling framework for learning complex high-dimensional distributions. Unlike likelihood-based methods, GANs are referred to as implicit probabilistic models (Mohamed & Lakshminarayanan, 2016). They represent a probability distribution through a generator that learns to directly produce samples from the desired distribution. The generator is trained adversarially by optimizing a minimax objective together with a discriminator. In practice, GANs have been shown to be very successful in a range of applications including generating photorealistic images (Karras et al., 2018). Other than generating samples, many downstream tasks require a good generative model, such as image inpainting (Pathak et al., 2016; Yeh et al., 2017).

Training GANs normally requires access to a large collection of fully-observed data. However, it is not always possible to obtain a large amount of fully-observed data. Missing data is well-known to be prevalent in many real-world application domains where different data cases might have different missing entries. This arbitrary missingness poses a significant challenge to many existing machine learning models.

Following Little & Rubin (2014), the generative process for incompletely observed data can be described as shown below where $\mathbf{x} \in \mathbb{R}^n$ is a complete data vector and $\mathbf{m} \in \{0, 1\}^n$ is a binary mask[2] that determines which entries in $\mathbf{x}$ to reveal:

$$\mathbf{x} \sim p_\theta(\mathbf{x}), \quad \mathbf{m} \sim p_\phi(\mathbf{m}|\mathbf{x}). \tag{1}$$

Let $\mathbf{x}_{\mathrm{obs}}$ denote the observed elements of $\mathbf{x}$, and $\mathbf{x}_{\mathrm{mis}}$ denote the missing elements according to the mask $\mathbf{m}$. In addition, let $\theta$ denote the unknown parameters of the data distribution, and $\phi$ denote the unknown parameters for the mask distribution, which are usually assumed to be independent of $\theta$. In the standard maximum likelihood setting, the unknown parameters are estimated by maximizing the

---

[1] Our implementation is available at https://github.com/steveli/misgan
[2] The complement $\bar{\mathbf{m}}$ is usually referred to as the missing data indicator in the literature.

following marginal likelihood, integrating over the unknown missing data values:

$$p(\mathbf{x}_{\text{obs}}, \mathbf{m}) = \int p_\theta(\mathbf{x}_{\text{obs}}, \mathbf{x}_{\text{mis}}) p_\phi(\mathbf{m}|\mathbf{x}_{\text{obs}}, \mathbf{x}_{\text{mis}}) d\mathbf{x}_{\text{mis}}.$$

Little & Rubin (2014) characterize the missing data mechanism $p_\phi(\mathbf{m}|\mathbf{x}_{\text{obs}}, \mathbf{x}_{\text{mis}})$ in terms of independence relations between the complete data $\mathbf{x} = [\mathbf{x}_{\text{obs}}, \mathbf{x}_{\text{mis}}]$ and the masks $\mathbf{m}$:

- Missing completely at random (MCAR): $p_\phi(\mathbf{m}|\mathbf{x}) = p_\phi(\mathbf{m})$,
- Missing at random (MAR): $p_\phi(\mathbf{m}|\mathbf{x}) = p_\phi(\mathbf{m}|\mathbf{x}_{\text{obs}})$,
- Not missing at random (NMAR): $\mathbf{m}$ depends on $\mathbf{x}_{\text{mis}}$ and possibly also $\mathbf{x}_{\text{obs}}$.

Most work on incomplete data assumes MCAR or MAR since under these assumptions $p(\mathbf{x}_{\text{obs}}, \mathbf{m})$ can be factorized into $p_\theta(\mathbf{x}_{\text{obs}}) p_\phi(\mathbf{m}|\mathbf{x}_{\text{obs}})$. With such decoupling, the missing data mechanism can be ignored when learning the data generating model while yielding correct estimates for $\theta$. When $p_\theta(\mathbf{x})$ does not admit efficient marginalization over $\mathbf{x}_{\text{mis}}$, estimation of $\theta$ is usually performed by maximizing a variational lower bound, as shown below, using the EM algorithm or a more general approach (Little & Rubin, 2014; Ghahramani & Jordan, 1994):

$$\log p_\theta(\mathbf{x}_{\text{obs}}) \geq \mathbb{E}_{q(\mathbf{x}_{\text{mis}}|\mathbf{x}_{\text{obs}})} \left[ \log p_\theta(\mathbf{x}_{\text{obs}}, \mathbf{x}_{\text{mis}}) - \log q(\mathbf{x}_{\text{mis}}|\mathbf{x}_{\text{obs}}) \right]. \tag{2}$$

The primary contribution of this paper is the development of a GAN-based framework for learning high-dimensional data distributions in the presence of incomplete observations. Our framework introduces an auxiliary GAN for learning a mask distribution to model the missingness. The masks are used to "mask" generated complete data by filling the indicated missing entries with a constant value. The complete data generator is trained so that the resulting masked data are indistinguishable from real incomplete data that are masked similarly.

Our framework builds on the ideas of AmbientGAN (Bora et al., 2018). AmbientGAN modifies the discriminator of a GAN to distinguish corrupted real samples from corrupted generated samples under a range of corruption processes (or measurement processes). For images, examples of the measurement processes include random dropout, blur, block-patch, and so on. Missing data can be seen as a special type of corruption, except that we have access to the missing pattern in addition to the corrupted measurements. Moreover, AmbientGAN assumes the measurement process is known or parameterized only by a few parameters, which is not the case in general missing data problems.

We provide empirical evidence that the proposed framework is able to effectively learn complex, high-dimensional data distributions from highly incomplete data when the GAN generator incorporates suitable priors on the data generating process. We further show how the architecture can be used to generate high-quality imputations.

## 2   MISGAN: A GAN FOR MISSING DATA

In the missing data problem, we know exactly which entries in each data examples are missing. Therefore, we can represent an incomplete data case as a pair of a partially-observed data vector $\mathbf{x} \in \mathbb{R}^n$ and a corresponding mask $\mathbf{m} \in \{0, 1\}^n$ that indicates which entries in $\mathbf{x}$ are observed: $x_d$ is observed if $m_d = 1$ otherwise $x_d$ is missing and might contain an arbitrary value that we should ignore. With this representation, an incomplete dataset is denoted $\mathcal{D} = \{(\mathbf{x}_i, \mathbf{m}_i)\}_{i=1,\ldots,N}$ (we assume instances are i.i.d. samples). We choose this representation instead of $\mathbf{x}_{\text{obs}}$ because it leads to a cleaner description of the proposed MisGAN framework. It also suggests how MisGAN can be implemented efficiently in practice as both $\mathbf{x}$ and $\mathbf{m}$ are fixed-length vectors.

We begin by defining a masking operator $f_\tau$ that fills in missing entries with a constant value $\tau$:

$$f_\tau(\mathbf{x}, \mathbf{m}) = \mathbf{x} \odot \mathbf{m} + \tau \bar{\mathbf{m}}, \tag{3}$$

where $\bar{\mathbf{m}}$ denotes the complement of $\mathbf{m}$ and $\odot$ denotes element-wise multiplication.

Two key ideas underlie the MisGAN framework. First, in addition to the complete data generator, we explicitly model the missing data process using a mask generator. Since the masks in the incomplete dataset are fully observed, we can estimate their distribution using a standard GAN. Second, we train

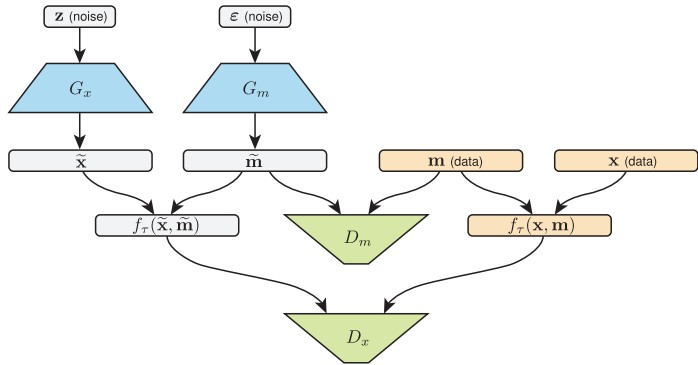

Figure 1: Overall structure of the MisGAN framework

the complete data generator adversarially by masking its outputs using generated masks and $f_\tau$ and comparing to real incomplete data that are similarly masked by $f_\tau$.

Specifically, we use two generator-discriminator pairs $(G_m, D_m)$ and $(G_x, D_x)$ for the masks and data respectively. In this paper, we focus on the missing completely at random (MCAR) case, where the two generators are independent of each other and have their own noise distributions $p_z$ and $p_\varepsilon$. We define the following two loss functions, one for the masks and the other for the data:

$$\mathcal{L}_m(D_m, G_m) = \mathbb{E}_{(\mathbf{x}, \mathbf{m}) \sim p_\mathcal{D}} \left[ D_m(\mathbf{m}) \right] - \mathbb{E}_{\boldsymbol{\varepsilon} \sim p_\varepsilon} \left[ D_m(G_m(\boldsymbol{\varepsilon})) \right], \tag{4}$$

$$\mathcal{L}_x(D_x, G_x, G_m) = \mathbb{E}_{(\mathbf{x}, \mathbf{m}) \sim p_\mathcal{D}} \left[ D_x(f_\tau(\mathbf{x}, \mathbf{m})) \right] - \mathbb{E}_{\boldsymbol{\varepsilon} \sim p_\varepsilon, \mathbf{z} \sim p_z} \left[ D_x \left( f_\tau \left( G_x(\mathbf{z}), G_m(\boldsymbol{\varepsilon}) \right) \right) \right]. \tag{5}$$

The losses above follow the Wasserstein GAN formulation (Arjovsky et al., 2017), although the proposed framework is compatible with many GAN variations (Goodfellow et al., 2014; Berthelot et al., 2017; Gulrajani et al., 2017). We optimize the generators and the discriminators according to the following objectives:

$$\min_{G_x} \max_{D_x \in \mathcal{F}_x} \mathcal{L}_x(D_x, G_x, G_m), \tag{6}$$

$$\min_{G_m} \max_{D_m \in \mathcal{F}_m} \mathcal{L}_m(D_m, G_m) + \alpha \mathcal{L}_x(D_x, G_x, G_m), \tag{7}$$

where $\mathcal{F}_x, \mathcal{F}_m$ are defined such that $D_x, D_m$ are both 1-Lipschitz for Wasserstein GANs (Arjovsky et al., 2017). Practically, we follow the common practice of alternating between a few steps of optimizing the discriminators and one step of optimizing the generators (Goodfellow et al., 2014; Arjovsky et al., 2017; Gulrajani et al., 2017). The coefficient $\alpha$ is introduced when optimizing the mask generator $G_m$ with the aim of minimizing a combination of $\mathcal{L}_m$ and $\mathcal{L}_x$. Although in theory we could choose $\alpha = 0$ to train $G_m$ and $D_m$ without using the data, we find that choosing a small value such as $\alpha = 0.2$ improves performance. This encourages the generated masks to match the distribution of the real masks and the masked generated complete samples to match masked real data. The overall structure of MisGAN is illustrated in Figure 1.

Note that the data discriminator $D_x$ takes as input the masked samples as if the data are fully-observed. This allows us to use any existing architecture designed for complete data to construct the data discriminator. There is no need to develop customized neural network modules for dealing with missing data. For example, $D_x$ can be a standard convolutional network for image applications.

Note that the masks are binary-valued. Since discrete data generating processes have zero gradient almost everywhere, to carry out gradient-based training for GANs, we relax the output of the mask generator $G_m$ from $\{0, 1\}^n$ to $[0, 1]^n$. We employ a sigmoid activation $\sigma_\lambda(x) = 1/(1 + \exp(-x/\lambda))$ with a low temperature $0 < \lambda < 1$ to encourage saturation and make the output closer to zero or one.

Finally, we note that the discriminator $D_x$ in MisGAN is unaware of which entries are missing in the masked input samples, and does not even need to know which value $\tau$ is used for masking. In the next section, we present a theoretical analysis providing support for the idea that this type of masking process does not necessarily make it more difficult to recover the complete data distribution. The experiments provide compelling empirical evidence for the effectiveness of the proposed framework.

## 3 THEORETICAL RESULTS

In Section 2 we described how the discriminator $D_x$ in MisGAN takes as input the masked samples using (3) without knowing what value $\tau$ is used or which entries in the input vector are missing. In this section, we discuss the following two important questions: i) Does the choice of the filled-in value $\tau$ affect the ability to recover the data distribution? ii) Does information about the location of missing values affect the ability to recover the data distribution?

We address these questions in a simplified scenario where each dimension of the data vector takes values from a finite set $\mathcal{P}$. For $n$-dimensional data, let $\mathcal{M} = \{0,1\}^n$ be the set of all possible masks and $\mathcal{I} = \mathcal{P}^n$ be the set of all possible data vectors. Also let $\mathcal{D}_{\mathcal{M}}$ and $\mathcal{D}_{\mathcal{I}}$ be the set of all possible probability distributions on $\mathcal{M}$ and $\mathcal{I}$ respectively, whose elements are non-negative and sum to one. We first discuss the case where the filled-in value $\tau$ is chosen from $\mathcal{P}$.

Given $\tau \in \mathcal{P}$ and $\mathbf{q} \in \mathcal{D}_{\mathcal{M}}$, we can construct a left transition matrix $T_{\mathbf{q},\tau} \in \mathbb{R}^{\mathcal{I} \times \mathcal{I}}$ defined below where the $(\mathbf{t}, \mathbf{s})$-th entry specifies the transition probability from a data vector $\mathbf{s} \in \mathcal{I}$ to an outcome $\mathbf{t} \in \mathcal{I}$ masked by $f_\tau$, which involves all possible masks under which $\mathbf{s}$ is converted into $\mathbf{t}$ by filling in the indicated missing entries with $\tau$:

$$T_{\mathbf{q},\tau}(\mathbf{t}, \mathbf{s}) = \sum_{\mathbf{m} \in \mathcal{M}: f_\tau(\mathbf{s}, \mathbf{m}) = \mathbf{t}} \mathbf{q}(\mathbf{m}).$$

Let $\mathbf{p}_x^* \in \mathcal{D}_{\mathcal{I}}$ be the unknown true data distribution we want to estimate. In the presence of missing data specified by $\mathbf{q}$, the masked samples then follow the distribution $\mathbf{p}_y = T_{\mathbf{q},\tau} \mathbf{p}_x^*$. Without imposing extra application-specific constraints, MisGAN with a fixed mask generator can be viewed as solving the linear system $\mathbf{p}_y = T_{\mathbf{q},\tau} \mathbf{p}_x$, where $\mathbf{p}_x \in \mathcal{D}_{\mathcal{I}}$ is the unknown data distribution to solve for. Here we assume that $\mathbf{p}_y$ and $T_{\mathbf{q},\tau}$ are given, as those can be estimated separately from a collection of fully-observed masks and masked samples.

Note that a transition matrix preserves the sum of the vectors it is applied to since $\mathbf{1}^\top T_{\mathbf{q},\tau} = \mathbf{1}^\top$. For $\mathbf{p}_x$ to be a valid distribution vector, we only need the non-negativity constraint because any solution $\mathbf{p}_x$ automatically sums to one. That is, estimating the data generating process in the presence of missing data based on the masking scheme used in MisGAN is equivalent to solving the linear system

$$T_{\mathbf{q},\tau} \mathbf{p}_x = \mathbf{p}_y \ \text{ subject to } \mathbf{p}_x \succeq \mathbf{0}. \tag{8}$$

In Theorem 1, we state a key property of the transition matrix $T_{\mathbf{q},\tau}$ that leads to the answer to our questions. The proof of Theorem 1 is in Appendix A.

**Theorem 1.** *Given $\mathbf{q} \in \mathcal{D}_{\mathcal{M}}$, all transition matrices $T_{\mathbf{q},\tau}$ with $\tau \in \mathcal{P}$ have the same null space.*

Theorem 1 implies that if the solution to the constrained linear system (8) is not unique for a given $\tau_0 \in \mathcal{P}$, that is, there exists some non-negative $\mathbf{p}_x \neq \mathbf{p}_x^*$ such that $T_{\mathbf{q},\tau_0} \mathbf{p}_x = T_{\mathbf{q},\tau_0} \mathbf{p}_x^*$, then we must have $T_{\mathbf{q},\tau} \mathbf{p}_x = T_{\mathbf{q},\tau} \mathbf{p}_x^*$ for all $\tau \in \mathcal{P}$. In other words, we have the following corollary:

**Corollary 1.** *Whether the true data distribution is uniquely recoverable is independent of the choice of the filled-in value $\tau$.*

Here we only discuss the case when the probability of observing all features $\mathbf{q}(\mathbf{1})$ is zero, where $\mathbf{q}(\mathbf{1})$ denotes the scalar entry of $\mathbf{q}$ indexed by $\mathbf{1} \in \mathcal{M}$. Otherwise, the linear system is uniquely solvable as the transition matrix $T_{\mathbf{q},\tau_0}$ has full rank. With the non-negativity constraint, it is possible that the solution for the linear system (8) is unique when the true data distribution $\mathbf{p}_x^*$ is sparse. Specifically, if there exists two indices $\mathbf{s}_1, \mathbf{s}_2 \in \mathcal{I}$ such that $\mathbf{p}_x^*(\mathbf{s}_1) = \mathbf{p}_x^*(\mathbf{s}_2) = 0$ and also $\mathbf{v}(\mathbf{s}_1) > 0$ and $\mathbf{v}(\mathbf{s}_2) < 0$ for all $\mathbf{v} \in \text{Null}(T_{\mathbf{q},\tau}) \setminus \{\mathbf{0}\}$, then the solution to (8) is unique.

Sparsity of the data distribution is a reasonable assumption in many situations. For example, natural images are typically considered to lie on a low dimensional manifold, which means most of the instances in $\mathcal{I}$ should have almost zero probability. On the other hand, when the missing rate is high, that is, if the masks in $\mathcal{M}$ that have many zeros are more probable, the null space of $T_{\mathbf{q},\tau}$ will be larger and therefore it is more likely that the non-negative solution is not unique. Bruckstein et al. (2008) proposed a sufficient condition on the sparsity of the non-negative solutions to a general underdetermined linear system that guarantees unique optimality.

Next we note that in the case of $\tau \in \mathcal{P}$, an entry with value $\tau$ in a masked sample $\mathbf{t} \in \mathcal{I}$ may come either from an observed entry with value $\tau$ in the unmasked sample or from an unobserved entry through the masking operation in (3). One might wonder if this prevents an algorithm from recovering the true distribution when it is otherwise possible to do so. In other words, if we take the location of the missing values into account, would that make the missing data problem less ill-posed? However, this is not the case, as we state in Corollary 2. The proof is in Appendix B where we discuss the case of $\tau \notin \mathcal{P}$.

**Corollary 2.** *If the linear system $T_{\mathbf{q},\tau} \mathbf{p}_x = T_{\mathbf{q},\tau} \mathbf{p}_x^*$ does not have a unique non-negative solution, then for this missing data problem, we cannot uniquely recover the true data distribution even if we take the location of the missing values into account.*

Note that the analysis in this section characterizes how difficult the missing data problem is, which is independent of the choice of the algorithm that solves it. In practice, it is useful to incorporate application-specific prior knowledge into the model to regularize the problem when it is ill-posed. For example, for modeling natural images, convolutional networks are commonly used to exploit the local structure of the data. In addition, decoder-based deep generative models such as GANs implicitly enforce some sparsity constraints due to the use of low dimensional latent codes in the generator, which also helps to regularize the problem.

Finally, the following theorem justifies the training objective (6) of MisGAN for the missing data problem (see Appendix A for details).

**Theorem 2.** *Given a mask distribution $p_\phi(\mathbf{m})$, two distributions $p_\theta(\mathbf{x})$ and $p_{\theta'}(\mathbf{x})$ induce the same distribution for $f_\tau(\mathbf{x}, \mathbf{m})$ if and only if they have the same marginals $p_\theta(\mathbf{x}_{\text{obs}}|\mathbf{m}) = p_{\theta'}(\mathbf{x}_{\text{obs}}|\mathbf{m})$ for all masks $\mathbf{m}$ with $p_\phi(\mathbf{m}) > 0$.*[3]

## 4 MISSING DATA IMPUTATION

Missing data imputation is an important task when dealing with incomplete data. In this section, we show how to impute missing data according to $p(\mathbf{x}_{\text{mis}}|\mathbf{x}_{\text{obs}})$ by equipping MisGAN with an imputer $G_i$ accompanied by a corresponding discriminator $D_i$. The imputer is a function of the incomplete example $(\mathbf{x}, \mathbf{m})$ and a random vector $\boldsymbol{\omega}$ drawn from a noise distribution $p_\omega$. It outputs the completed sample with the observed part in $\mathbf{x}$ kept intact.

To train the imputer-equipped MisGAN, we define the loss for the imputer in addition to (4) and (5):

$$\mathcal{L}_i(D_i, G_i, G_x) = \mathbb{E}_{\mathbf{z} \sim p_z} [D_i(G_x(\mathbf{z}))] - \mathbb{E}_{(\mathbf{x},\mathbf{m}) \sim p_\mathcal{D}, \boldsymbol{\omega} \sim p_\omega} [D_i(G_i(\mathbf{x}, \mathbf{m}, \boldsymbol{\omega}))].$$

We jointly learn the data generating process and the imputer according to the following objectives:

$$\min_{G_i} \max_{D_i \in \mathcal{F}_i} \mathcal{L}_i(D_i, G_i, G_x), \tag{9}$$

$$\min_{G_x} \max_{D_x \in \mathcal{F}_x} \mathcal{L}_x(D_x, G_x, G_m) + \beta \mathcal{L}_i(D_i, G_i, G_x), \tag{10}$$

$$\min_{G_m} \max_{D_m \in \mathcal{F}_m} \mathcal{L}_m(D_m, G_m) + \alpha \mathcal{L}_x(D_x, G_x, G_m),$$

where we use $\beta = 0.1$ in the experiments when optimizing $G_x$. This encourages the generated complete data to match the distribution of the imputed real data in addition to having the masked generated data match the masked real data. The overall structure for MisGAN imputation is illustrated in Figure 2.

We can also train a stand-alone imputer using only (9) with a pre-trained data generator $G_x$. The architecture is as shown in Figure 2 with the faded parts removed. Moreover, it is also possible to train the imputer to target a different missing distribution $p_m$ with a pre-trained data generator $G_x$ alone without access to the original (incomplete) training data:

$$\min_{G_i} \max_{D_i \in \mathcal{F}_i} \mathbb{E}_{\mathbf{z} \sim p_z} [D_i(G_x(\mathbf{z}))] - \mathbb{E}_{\mathbf{m} \sim p_m, \mathbf{z} \sim p_z, \boldsymbol{\omega} \sim p_\omega} [D_i(G_i(G_x(\mathbf{z}), \mathbf{m}, \boldsymbol{\omega}))]. \tag{11}$$

We construct the imputer $G_i(\mathbf{x}, \mathbf{m}, \boldsymbol{\omega})$ as follows:

$$G_i(\mathbf{x}, \mathbf{m}, \boldsymbol{\omega}) = \mathbf{x} \odot \mathbf{m} + \widehat{G}_i(\mathbf{x} \odot \mathbf{m} + \boldsymbol{\omega} \odot \bar{\mathbf{m}}) \odot \bar{\mathbf{m}}, \tag{12}$$

---

[3] $p_\theta(\mathbf{x}_{\text{obs}}|\mathbf{m})$ is technically equivalent to $p_\theta(\mathbf{x}_{\text{obs}})$ as the random variable $\mathbf{x}_{\text{obs}} = \{x_d : m_d = 1\}$ is defined with a known mask $\mathbf{m}$.

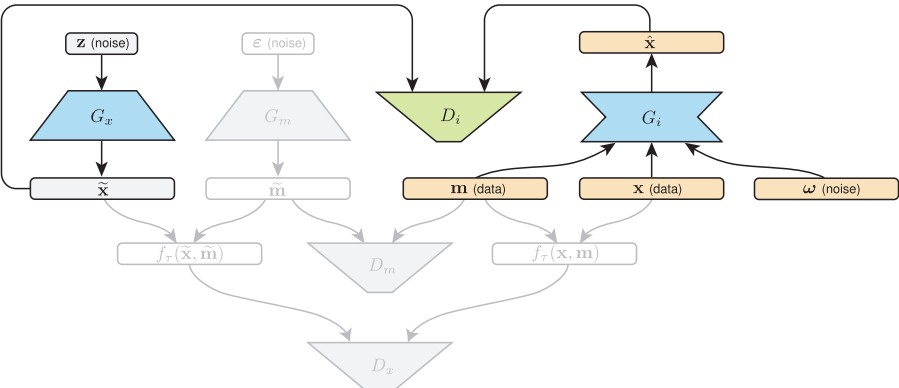

Figure 2: Architecture for MisGAN imputation. The complete data generator $G_x$ and the imputer $G_i$ can be trained jointly with all the components. We can also independently train the imputer $G_i$ without the faded parts if the data generator $G_x$ has been pre-trained.

where $\widehat{G}_i$ generates the imputed result with the same dimensionality as its input, $\mathbf{x} \odot \mathbf{m} + \boldsymbol{\omega} \odot \bar{\mathbf{m}}$, which could be implemented by a deep neural network. The masking outside of $\widehat{G}_i$ ensures that the observed part of $\mathbf{x}$ stays the same in the output of the imputer $G_i$. The similar masking on the input of $\widehat{G}_i$, $\mathbf{x} \odot \mathbf{m} + \boldsymbol{\omega} \odot \bar{\mathbf{m}}$, ensures that the amount of noise injected to $\widehat{G}_i$ scales with the number of missing dimensions. This is intuitive in the sense that when a data case is almost fully-observed, we expect less variety in $p(\mathbf{x}_{\mathrm{mis}}|\mathbf{x}_{\mathrm{obs}})$ and vice versa. Note that the noise $\boldsymbol{\omega}$ needs to have the same dimensionality as $\mathbf{x}$.

## 5 EXPERIMENTS

In this section, we first assess various properties of MisGAN on the MNIST dataset: we demonstrate qualitatively how MisGAN behaves under different missing patterns and different architectures. We then conduct an ablation study to justify the construction of MisGAN. Finally, we compare MisGAN with various baseline methods on the missing data imputation task over three datasets under a series of missingness settings.

**Data** We evaluate MisGAN on three datasets: MNIST, CIFAR-10 and CelebA. MNIST is a dataset of handwritten digits images of size $28 \times 28$ (LeCun et al., 1998). We use the provided 60,000 training examples for the experiments. CIFAR-10 is a dataset of $32 \times 32$ color images from 10 classes (Krizhevsky, 2009). Similarly, we use 50,000 training examples for the experiments. CelebA is a large-scale face attributes dataset (Liu et al., 2015) that contains 202,599 face images, where we use the provided aligned and cropped images and resize them to $64 \times 64$. For all three datasets, the range of pixel values of each image is rescaled to $[0, 1]$.

**Missing data distributions** We consider three types of missing data distribution: i) *Square observation*: all pixels are missing except for a square occurring at a random location on the image. ii) *Dropout*: each pixel is independently missing according to a Bernoulli distribution. iii) *Variable–size rectangular observation*: all pixels are missing except for a rectangular observed region. The width and height of the rectangle are independently drawn from 25% to 75% of the image length uniformly at random, which results in a 75% missing rate on average. In this missing data distribution, each example may have a different number of missing pixels. The highest per-example missing data rate under this mechanism is 93.75%.

**Evaluation metric** We use the Fréchet Inception Distance (FID) (Heusel et al., 2017) to evaluate the quality of the learned generative model. For MNIST, instead of the Inception network trained on ImageNet (Salimans et al., 2016), we use a basic LeNet model [4] trained on the complete MNIST training set, and then take the 50-dimensional output from the second-to-last fully-connected layer as the features to compute the FID. For CIFAR-10 and CelebA, we follow the procedure described in

---

[4] https://github.com/pytorch/examples/tree/master/mnist

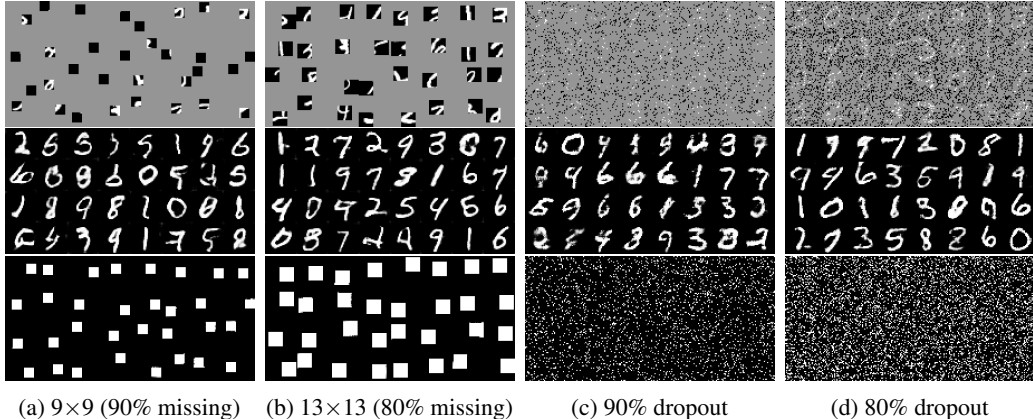

(a) 9×9 (90% missing)   (b) 13×13 (80% missing)   (c) 90% dropout   (d) 80% dropout

Figure 3: Conv-MisGAN results under different missing data processes. Top: training samples where gray pixels indicate missing data. Middle: data samples generated by $G_x$. Bottom: mask samples generated by $G_m$.

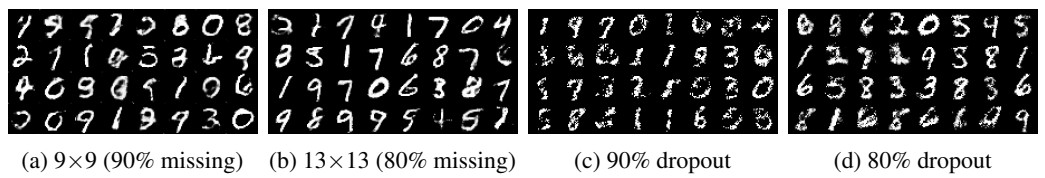

(a) 9×9 (90% missing)   (b) 13×13 (80% missing)   (c) 90% dropout   (d) 80% dropout

Figure 4: Data samples generated by FC-MisGAN.

Heusel et al. (2017) to compute the FID using the pretrained Inception-v3 model. When evaluating generative models using the FID, we use the same number of generated samples as the size of the training set.

## 5.1 EMPIRICAL STUDY OF MISGAN ON MNIST

In this section, we study various properties of MisGAN using the MNIST dataset.

**Architectures**  We consider two kinds of architecture for MisGAN: convolutional networks and fully connected networks. We follow the DCGAN architecture (Radford et al., 2015) for (de)convolutional generators and discriminators to exploit the local structures of images. We call this model Conv-MisGAN.

To demonstrate the performance of MisGAN in the absence of the implicit structural regularization provided by the use of a convolutional network, we construct another MisGAN with only fully-connected layers for both the generators and the discriminators, which we call FC-MisGAN.

In the experiments, both Conv-MisGAN and FC-MisGAN are trained using the improved procedure for the Wasserstein GAN with gradient penalty (Gulrajani et al., 2017). Throughout we use $\tau = 0$ for the masking operator and the temperature $\lambda = 0.66$ for the mask activation $\sigma_\lambda(x)$ described in Section 2.

**Baseline**  We compare MisGAN to a baseline model that is capable of learning from large-scale incomplete data: the generative convolutional arithmetic circuit (ConvAC) (Sharir et al., 2016). ConvAC is an expressive mixture model similar to sum-product networks (Poon & Domingos, 2011) with a compositional structure similar to deep convolutional networks. Most importantly, ConvAC admits tractable marginalization due to the product form of the base distributions for the mixtures, which makes it readily capable of learning with missing data.

**Results**  Figures 3 and 4 show the generated data samples as well as the learned mask samples produced by Conv-MisGAN and FC-MisGAN under the square observation and independent dropout missing mechanisms. From these results, we can see that Conv-MisGAN produces visually better

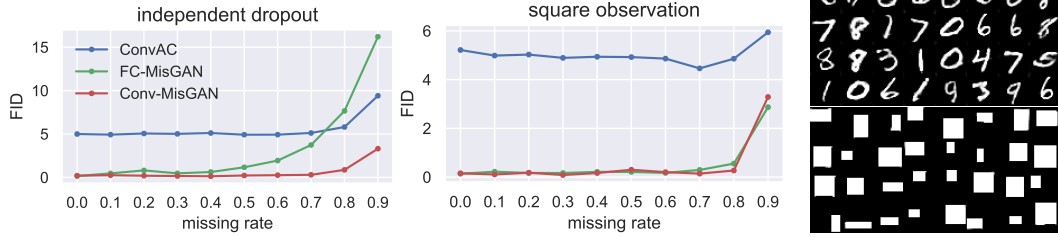

Figure 5: Left & Middle: Missing rate versus FID (The lower the better) with different missing data processes. Right: Data samples (top) and mask samples (bottom) generated by Conv-MisGAN learned with variable-size observations.

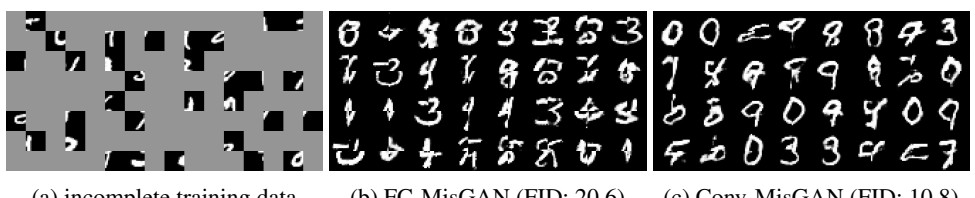

(a) incomplete training data     (b) FC-MisGAN (FID: 20.6)     (c) Conv-MisGAN (FID: 10.8)

Figure 6: Data samples generated by MisGAN when trained on missing data distributions with non-overlapping samples (square quadrants).

samples than FC-MisGAN on this problem. On the other hand, under the same missing rate, independent dropout leads to worse samples than square observations. Samples generated by ConvAC are shown in Figure 18 in Appendix G.

We quantitatively evaluate Conv-MisGAN, FC-MisGAN and ConvAC under two missing patterns with missing rates from 10% to 90% with a step of 10%. Figure 5 shows that MisGAN in general outperforms ConvAC as ConvAC tends to generate samples with aliasing artifacts as shown in Figure 18. It also shows that in the square observation case, Conv-MisGAN and FC-MisGAN have similar performance in terms of their FIDs. However, under independent dropout, the performance of FC-MisGAN degrades significantly as the missing rate increases compared to Conv-MisGAN. This is because independent dropout with high missing rate makes the problem more challenging as it induces less overlapping co-occurrence among pixels, which degrades the signal for understanding the overall structure.

This is illustrated in Figure 6 where the observed pattern comes from one of four equally probable 14×14 square quadrants with no overlap. Clearly this missing data problem is ill-posed and we could never uniquely determine the correlation between pixels across different quadrants without additional assumptions. The samples generated by the FC-MisGAN produce obvious discontinuity across the boundary of the quadrants as it does not rely on any prior knowledge about how pixels are correlated. The discontinuity artifact is less severe with Conv-MisGAN since the convolutional layers encourage local smoothness. This shows the importance of incorporating prior knowledge into the model when the problem is highly ill-posed.

**Ablation study**   We point out that the mask discriminator in MisGAN is important for learning the correct distribution robustly. Figure 7 shows two common failure scenarios that frequently happen with an AmbientGAN, which is essentially equivalent to a MisGAN without the mask discriminator. Figure 7 (left) shows a case where AmbientGAN produces perfectly consistent masked outputs, but the learned mask distribution is completely wrong. Since we use $f_{\tau=0}(\mathbf{x}, \mathbf{m}) = \mathbf{x} \odot \mathbf{m}$, it makes the role of $\mathbf{x}$ and $\mathbf{m}$ interchangeable when considering only the masked outputs. Even if we rescale the range of pixel values from $[0, 1]$ to $[-1, 1]$ to avoid this situation, AmbientGAN still fails often as shown in Figure 7 (right). In contrast, MisGAN avoids learning such degenerate solutions due to explicitly modeling the mask distribution.

**Missing data imputation**   We construct the imputer network $\widehat{G}_i$ defined in (12) using a three-layer fully-connected network with 500 hidden units in the middle layers. Figure 8 (left) shows the

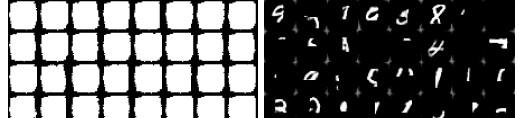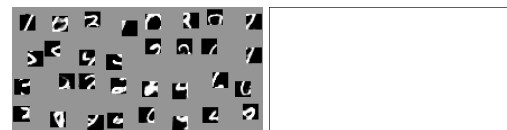

Figure 7: Two failure cases of AmbientGAN. In each pair, data samples produced by $G_x$ are on the left, mask samples from $G_m$ are on the right. In the right panels, the range of pixel values is rescaled to $[-1, 1]$ so gray pixels correspond to $\tau = 0$. It learns the masks with all ones.

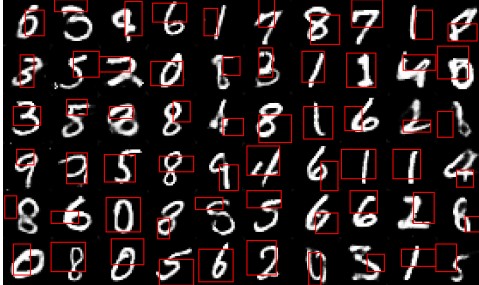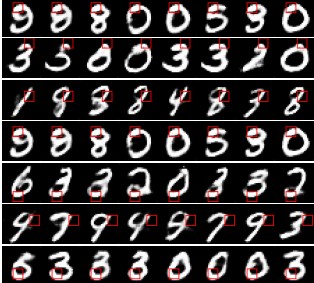

Figure 8: Inside of each red box are the observed pixels; the pixels outside of the box are generated by the imputer. Right: each row corresponds to the same incomplete input, marked by the red box.

imputation results on different examples applying novel masks randomly drawn according to the same distribution. Figure 8 (right) shows the imputation results where each row corresponds to the same incomplete input. It demonstrates that the imputer can produce a variety of different imputed results due to the random noise input to the imputer. We also note that if we modify (11) to train the imputer together with the data generator from scratch without the mask generator/discriminator, it fails most of the time for a similar reason to why AmbientGAN fails. The learning problem is highly ill-posed without the agreement on the mask distribution.

## 5.2 QUANTITATIVE EVALUATION

In this section, we quantitatively evaluate the performance of MisGAN on three datasets: MNIST, CIFAR-10, and CelebA. We focus on evaluating MisGAN on the missing data imputation task as it is widely studied and many baseline methods are available.

**Baselines**  We compare the MisGAN imputer to a range of baseline methods including the basic zero/mean imputation, matrix factorization, and the recently proposed Generative Adversarial Imputation Network (GAIN) (Yoon et al., 2018). GAIN is an imputation model that employs an imputer network to complete the missing data. It is trained adversarially with a discriminator that determines which entries in the completed data were actually observed and which were imputed. It has shown to outperform many state-of-the-art imputation methods.

**Evaluation of imputation**  We impute all of the incomplete examples in the training set and use the FID between the imputed data and the original fully-observed data as the evaluation metric.[5]

**Architecture**  We use convolutional generators and discriminators for MisGAN for all experiments in this section. For MNIST, we use the same fully-connected imputer network as described in the previous section; for CIFAR-10 and CelebA, we use a five-layer U-Net architecture (Ronneberger et al., 2015) for the imputer network $\widehat{G}_i$ in MisGAN.

**Results**  We compare all the methods under two missing patterns, square observation and independent dropout, with missing rates from 10% to 90%. Figure 9 shows that MisGAN consistently outperforms other methods in all cases, especially under high missing rates. In our experiments, we found GAIN training to be quite unstable for the block missingness. We also observed that there is a "sweet spot" for the number of training epochs when training GAIN. If trained longer, the imputation behavior will

---

[5] See Appendix C for a discussion of why we favor this metric over evaluating metrics like RMSE between the imputed missing values and the ground truth.

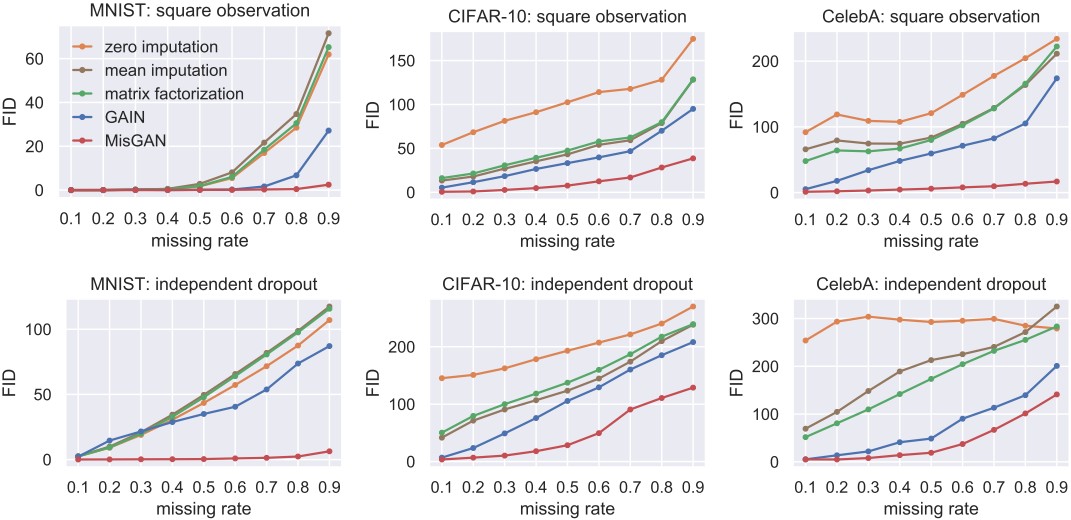

Figure 9: Comparison of FID across different missing rates.

gradually become similar to constant imputation (see Appendix H for details). On the other hand, we find that training MisGAN is more stable than training GAIN across all scenarios in the experiments. The imputation results of MisGAN and GAIN are shown in Appendix E, F, and H.

## 6 DISCUSSION AND FUTURE WORK

This work presents and evaluates a highly flexible framework for learning standard GAN data generators in the presence of missing data. Although we only focus on the MCAR case in this work, MisGAN can be easily extended to cases where the output of the data generator is provided to the mask generator. These modifications can capture both MAR and NMAR mechanisms. The question of learnability requires further investigation as the analysis in Section 3 no longer holds due to dependence between the transition matrix and the data distribution under MAR and NMAR. We have tried this modified architecture in our experiments and it showed similar results as to the original MisGAN. This suggests that the extra dependencies may not adversely affect learnability. We leave the formal evaluation of this modified framework for future work.

## ACKNOWLEDGEMENTS

This work was supported by the National Science Foundation under Grant No. IIS-1350522.

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

# A  PROOF OF THEOREM 1 AND THEOREM 2

Let $\mathcal{P}$ be the finite set of feature values. For the $n$-dimensional case, let $\mathcal{M} = \{0, 1\}^n$ be the set of masks and $\mathcal{I} = \mathcal{P}^n$ be the set of all possible feature vectors. Also let $\mathcal{D}_\mathcal{M}$ be the set of probability distributions on $\mathcal{M}$, which implies $\mathbf{m} \succeq \mathbf{0}$ and $\sum_{\mathbf{v} \in \mathcal{I}} \mathbf{m}(\mathbf{v}) = 1$ for all $\mathbf{m} \in \mathcal{M}$, where $\mathbf{m}(\mathbf{v})$ denotes the entry of $\mathbf{m}$ indexed by $\mathbf{v}$.

Given $\tau \in \mathcal{P}$ and $\mathbf{q} \in \mathcal{D}_\mathcal{M}$, define the transformation

$$T_{\mathbf{q},\tau} : \mathbb{R}^\mathcal{I} \to \mathbb{R}^\mathcal{I}$$
$$\mathbf{x} \mapsto \mathbf{y} = T_{\mathbf{q},\tau} \mathbf{x}$$

by

$$\mathbf{y}(\mathbf{v}) = (T_{\mathbf{q},\tau}\mathbf{x})(\mathbf{v}) = \sum_{\mathbf{m} \in \mathcal{M}} \sum_{\mathbf{u} \in \mathcal{I}} \mathbf{q}(\mathbf{m})\mathbf{x}(\mathbf{u})\mathbf{1}\{\mathbf{u} \odot \mathbf{m} + \tau\bar{\mathbf{m}} = \mathbf{v}\}, \quad \text{for all } \mathbf{v} \in \mathcal{I} \quad (13)$$

where $\odot$ is the entry-wise multiplication and $\mathbf{1}\{\cdot\}$ is the indicator function.

Given $\mathbf{m} \in \mathcal{M}$, define an equivalent relation $\sim_{\mathbf{m}}$ on $\mathcal{I}$ by $\mathbf{v} \sim_{\mathbf{m}} \mathbf{u}$ iff $\mathbf{v} \odot \mathbf{m} = \mathbf{u} \odot \mathbf{m}$, and denote by $[\mathbf{v}]_{\mathbf{m}}$ the equivalence class containing $\mathbf{v}$.

Given $\mathbf{q} \in \mathcal{D}_\mathcal{M}$, let $\mathcal{S}_{\mathbf{q}} \subset \mathcal{M}$ be the support of $\mathbf{q}$, that is,

$$\mathcal{S}_{\mathbf{q}} = \{\mathbf{m} \in \mathcal{M} : \mathbf{q}(\mathbf{m}) > 0\}.$$

Given $\tau \in \mathcal{P}$ and $\mathbf{v} \in \mathcal{I}$, let $\mathcal{M}_{\tau,\mathbf{v}}$ denote the set of masks consistent with $\mathbf{v}$ in the sense that $\mathbf{q}(\mathbf{m}) > 0$ and $\mathbf{v} \odot \bar{\mathbf{m}} = \tau\bar{\mathbf{m}}$, that is,

$$\mathcal{M}_{\tau,\mathbf{v}} = \{\mathbf{m} \in \mathcal{S}_{\mathbf{q}} : \mathbf{v} \odot \bar{\mathbf{m}} = \tau\bar{\mathbf{m}}\}.$$

**Proposition 1.** *For any $\mathbf{q} \in \mathcal{D}_\mathcal{M}$ and $\mathbf{x} \in \mathbb{R}^\mathcal{I}$, the collection of marginals $\{\mathbf{x}([\mathbf{v}]_{\mathbf{m}}) : \mathbf{v} \in \mathcal{I}, \mathbf{m} \in \mathcal{S}_{\mathbf{q}}\}$ determines $T_{\mathbf{q},\tau}\mathbf{x}$ for all $\tau \in \mathcal{P}$ where $\mathbf{x}([\mathbf{v}]_{\mathbf{m}}) := \sum_{\mathbf{u} \in [\mathbf{v}]_{\mathbf{m}}} \mathbf{x}(\mathbf{u})$.*

*Proof.* This is clear from the following equation

$$T_{\mathbf{q},\tau}\mathbf{x}(\mathbf{v}) = \sum_{\mathbf{m} \in \mathcal{M}_{\tau,\mathbf{v}}} \mathbf{q}(\mathbf{m})\mathbf{x}([\mathbf{v}]_{\mathbf{m}}), \quad (14)$$

which can be obtained from (13) as follows,

$$T_{\mathbf{q},\tau}\mathbf{x}(\mathbf{v}) = \sum_{\mathbf{m} \in \mathcal{S}_{\mathbf{q}}} \sum_{\mathbf{u} \in \mathcal{I}} \mathbf{q}(\mathbf{m})\mathbf{x}(\mathbf{u})\mathbf{1}\{\mathbf{u} \odot \mathbf{m} = \mathbf{v} \odot \mathbf{m}\}\mathbf{1}\{\tau\bar{\mathbf{m}} = \mathbf{v} \odot \bar{\mathbf{m}}\}$$
$$= \sum_{\mathbf{m} \in \mathcal{S}_{\mathbf{q}}} \mathbf{q}(\mathbf{m})\mathbf{1}\{\tau\bar{\mathbf{m}} = \mathbf{v} \odot \bar{\mathbf{m}}\} \sum_{\mathbf{u} \in \mathcal{I}} \mathbf{x}(\mathbf{u})\mathbf{1}\{\mathbf{u} \odot \mathbf{m} = \mathbf{v} \odot \mathbf{m}\}$$
$$= \sum_{\mathbf{m} \in \mathcal{S}_{\mathbf{q}}} \mathbf{q}(\mathbf{m})\mathbf{1}\{\tau\bar{\mathbf{m}} = \mathbf{v} \odot \bar{\mathbf{m}}\}\mathbf{x}([\mathbf{v}]_{\mathbf{m}})$$
$$= \sum_{\mathbf{m} \in \mathcal{M}_{\tau,\mathbf{v}}} \mathbf{q}(\mathbf{m})\mathbf{x}([\mathbf{v}]_{\mathbf{m}}).$$

$\square$

**Proposition 2.** *For any $\tau \in \mathcal{P}$, $\mathbf{q} \in \mathcal{D}_\mathcal{M}$ and $\mathbf{x} \in \mathbb{R}^\mathcal{I}$, the vector $T_{\mathbf{q},\tau}\mathbf{x}$ determines the collection of marginals $\{\mathbf{x}([\mathbf{v}]_{\mathbf{m}}) : \mathbf{v} \in \mathcal{I}, \mathbf{m} \in \mathcal{S}_{\mathbf{q}}\}$.*

*Proof.* Fix $\tau \in \mathcal{P}$, $\mathbf{q} \in \mathcal{D}_\mathcal{M}$ and $\mathbf{x} \in \mathbb{R}^\mathcal{I}$. Since $\mathbf{v} \odot \mathbf{m} + \tau\bar{\mathbf{m}} \in [\mathbf{v}]_{\mathbf{m}}$, it suffices to show that we can solve for $\mathbf{x}([\mathbf{v}]_{\mathbf{m}})$ in terms of $T_{\mathbf{q},\tau}\mathbf{x}$ for $\mathbf{m} \in \mathcal{M}_{\tau,\mathbf{v}} \neq \emptyset$. We use induction on the size of $\mathcal{M}_{\tau,\mathbf{v}}$.

First consider the base case $|\mathcal{M}_{\tau,\mathbf{v}}| = 1$. Consider $\mathbf{v}_0 \in \mathcal{I}$ with $\mathcal{M}_{\tau,\mathbf{v}_0} = \{\mathbf{m}_0\}$. By (14),

$$T_{\mathbf{q},\tau}\mathbf{x}(\mathbf{v}_0) = \mathbf{q}(\mathbf{m}_0)\mathbf{x}([\mathbf{v}_0]_{\mathbf{m}_0}).$$

Hence $\mathbf{x}([\mathbf{v}_0]_{\mathbf{m}_0}) = T_{\mathbf{q},\tau}\mathbf{x}(\mathbf{v}_0)/\mathbf{q}(\mathbf{m}_0)$, which proves the base case.

Now assume we can solve for $\mathbf{x}([\mathbf{v}]_{\mathbf{m}})$ in terms of $T_{\mathbf{q},\tau}\mathbf{x}$ for $\mathbf{m} \in \mathcal{S}_{\mathbf{q}}$ and $\mathbf{v} \in \mathcal{I}$ with $|\mathcal{M}_{\tau,\mathbf{v}}| \leq k$. Consider $\mathbf{v}_0 \in \mathcal{I}$ with $|\mathcal{M}_{\tau,\mathbf{v}_0}| = k + 1$; if no such $\mathbf{v}_0$ exists, the conclusion holds trivially. Let $\mathcal{M}_{\tau,\mathbf{v}_0} = \{\mathbf{m}_0, \mathbf{m}_1, \ldots, \mathbf{m}_k\}$. We need to show that $T_{\mathbf{q},\tau}\mathbf{x}$ determines $\mathbf{x}([\mathbf{v}_0]_{\mathbf{m}_\ell})$ for $\ell = 0, 1, \ldots, k$. By (14) again,

$$T_{\mathbf{q},\tau}\mathbf{x}(\mathbf{v}_0) = \sum_{\ell=0}^{k} \mathbf{q}(\mathbf{m}_\ell)\mathbf{x}([\mathbf{v}_0]_{\mathbf{m}_\ell}). \tag{15}$$

Let $\mathbf{m} = \bigwedge_{\ell=0}^{k} \mathbf{m}_\ell$, which may or may not belong to $\mathcal{S}_{\mathbf{q}}$. Note that

$$\mathbf{x}([\mathbf{v}_0]_{\mathbf{m}}) = \sum_{\mathbf{v} \in [\mathbf{v}_0]_{\mathbf{m} \vee \bar{\mathbf{m}}_\ell}} \mathbf{x}([\mathbf{v}]_{\mathbf{m}_\ell}) = \mathbf{x}([\mathbf{v}_0]_{\mathbf{m}_\ell}) + \sum_{\mathbf{v} \in [\mathbf{v}_0]_{\mathbf{m} \vee \bar{\mathbf{m}}_\ell} \setminus \{\mathbf{v}_0\}} \mathbf{x}([\mathbf{v}]_{\mathbf{m}_\ell}),$$

and hence

$$\mathbf{x}([\mathbf{v}_0]_{\mathbf{m}_\ell}) = \mathbf{x}([\mathbf{v}_0]_{\mathbf{m}}) - \sum_{\mathbf{v} \in [\mathbf{v}_0]_{\mathbf{m} \vee \bar{\mathbf{m}}_\ell} \setminus \{\mathbf{v}_0\}} \mathbf{x}([\mathbf{v}]_{\mathbf{m}_\ell}). \tag{16}$$

Plugging (16) into (15) yields

$$\mathbf{x}([\mathbf{v}_0]_{\mathbf{m}}) = \frac{1}{\sum_{\ell'=0}^{k} \mathbf{q}(\mathbf{m}_{\ell'})} \left[ T_{\mathbf{q},\tau}\mathbf{x}(\mathbf{v}_0) + \sum_{\ell=0}^{k} \mathbf{q}(\mathbf{m}_\ell) \sum_{\mathbf{v} \in [\mathbf{v}_0]_{\mathbf{m} \vee \bar{\mathbf{m}}_\ell} \setminus \{\mathbf{v}_0\}} \mathbf{x}([\mathbf{v}]_{\mathbf{m}_\ell}) \right]. \tag{17}$$

Note that $\mathcal{M}_{\tau,\mathbf{v}} \subset \mathcal{M}_{\tau,\mathbf{v}_0} \setminus \{\mathbf{m}_\ell\}$ for $\mathbf{v} \in [\mathbf{v}_0]_{\mathbf{m} \vee \bar{\mathbf{m}}_\ell} \setminus \{\mathbf{v}_0\}$, so $|\mathcal{M}_{\tau,\mathbf{v}}| \leq k$. By the induction hypothesis, $\mathbf{x}([\mathbf{v}]_{\mathbf{m}_\ell})$ is determined by $T_{\mathbf{q},\tau}\mathbf{x}$. It follows from (17) and (16) that $\mathbf{x}([\mathbf{v}_0]_{\mathbf{m}})$ and $\mathbf{x}([\mathbf{v}_0]_{\mathbf{m}_\ell})$ are also determined by $T_{\mathbf{q},\tau}\mathbf{x}$. This completes the induction step. $\square$

Theorem 1 is a direct consequence of Proposition 1 and Proposition 2 as the collection of marginals $\{\mathbf{x}([\mathbf{v}]_{\mathbf{m}}) : \mathbf{v} \in \mathcal{I}, \mathbf{m} \in \mathcal{S}_{\mathbf{q}}\}$ is independent of $\tau$. Therefore, if $\mathbf{x}_1, \mathbf{x}_2 \in \mathbb{R}^{\mathcal{I}}$ satisfy $T_{\mathbf{q},\tau_0}\mathbf{x}_1 = T_{\mathbf{q},\tau_0}\mathbf{x}_2$ for some $\tau_0 \in \mathcal{P}$, then $T_{\mathbf{q},\tau}\mathbf{x}_1 = T_{\mathbf{q},\tau}\mathbf{x}_2$ for all $\tau \in \mathcal{P}$. Theorem 1 is a special case when $\mathbf{x}_1 = \mathbf{0}$.

Moreover, Proposition 2 also shows that MisGAN overall learns the distribution $p(\mathbf{x}_{\text{obs}}, \mathbf{m})$, as $\mathbf{x}([\mathbf{v}]_{\mathbf{m}})$ is equivalent to $p(\mathbf{x}_{\text{obs}}|\mathbf{m})$ and $T_{\mathbf{q},\tau}\mathbf{x}$ is essentially the distribution of $f_\tau(\mathbf{x}, \mathbf{m})$ under the optimally learned missingness $\mathbf{q} = p(\mathbf{m})$. Theorem 2 basically restates Proposition 1 and Proposition 2. This is also true when $\tau \notin \mathcal{P}$ according to Appendix B.

## B  PROOF OF COROLLARY 2

Corollary 2 can be shown by augmenting the set of feature values by $\mathcal{P}' = \mathcal{P} \cup \{\psi\}$ with a novel symbol $\psi \notin \mathcal{P}$. If we choose $\tau = \psi$ for the masking operator, whenever we spot a $\psi$ in a masked sample, we know that it corresponds to a missing entry. We can also construct the corresponding transition matrix $T'_{\mathbf{q},\psi} \in \mathbb{R}^{\mathcal{I}' \times \mathcal{I}'}$ where $\mathcal{I}' = (\mathcal{P}')^n$ given the mask distribution $\mathbf{q} \in \mathcal{D}_{\mathcal{M}}$ before. In this setting, the generative model for missing data is equivalent to solving the linear system $T_{\mathbf{q},\psi}\mathbf{p}'_x = T_{\mathbf{q},\psi}\mathbf{p}'^*_x$ so that $\mathbf{p}'_x \in \mathbb{R}^{\mathcal{I}'}$ is non-negative and $\mathbf{p}'_x(\mathbf{s}) = 0$ for all $\mathbf{s} \in \mathcal{I}' \setminus \mathcal{I}$, where the true distribution $\mathbf{p}'^*_x$ is given by $\mathbf{p}'^*_x(\mathbf{s}) = \mathbf{p}^*_x(\mathbf{s})$ for all $\mathbf{s} \in \mathcal{I}$ and zeros elsewhere. Theorem 1 implies that if the solution to original problem (8) is not unique, the non-negative solution to the augmented linear system with the extra constraint on $\mathcal{I}' \setminus \mathcal{I}$ with $\tau = \psi$ is not unique either.

## C  EVALUATION OF IMPUTATION USING ROOT MEAN SQUARE ERROR

Root mean square error (RMSE) is a commonly used metric for evaluating the performance of missing data imputation, which computes the RMSE of the imputed missing values against the ground truth. However, in a complex system, the conditional distribution $p(\mathbf{x}_{\text{mis}}|\mathbf{x}_{\text{obs}})$ is likely to be highly multimodal. It's not guaranteed that the ground truth of the missing values in the incomplete dataset created under the missing completely at random (MCAR) assumption correspond to the global mode of $p(\mathbf{x}_{\text{mis}}|\mathbf{x}_{\text{obs}})$. A good imputation model might produce samples from $p(\mathbf{x}_{\text{mis}}|\mathbf{x}_{\text{obs}})$ associated with a higher density than the ground truth (or from other modes that are similarly probable). In this case, it will lead to a large error in terms of metrics like RMSE as multiple modes might be far

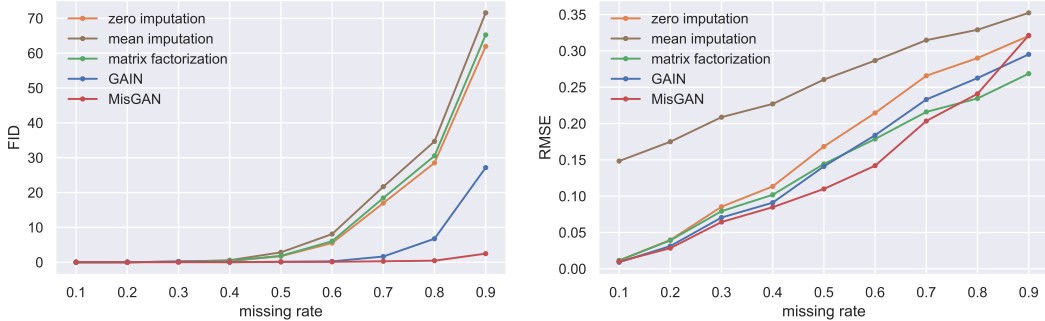

Figure 10: Comparison of evaluating imputation using FID and RMSE (both the lower the better) on the MNIST dataset with block observation missingness. The rankings of the imputation methods are not consistent across the two metrics under most of the assessed missing rates.

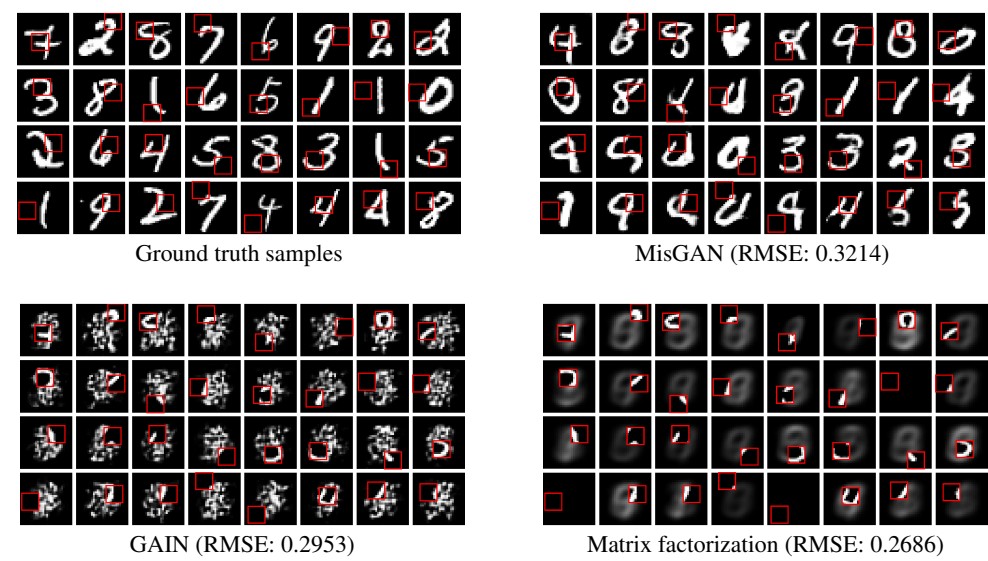

Figure 11: Imputation results by MisGAN, GAIN and matrix factorization along with the corresponding RMSE with block observation missingness under 90% missing rate. Inside of each red box are the observed pixels; the pixels outside of the box are generated by the imputation methods.

away from each other in a complex distribution. Therefore, we instead compute the FID between the distribution of the completed data and the distribution of the originally fully-observed data as our evaluation metric. This provides a practical way to assess how close a model imputes according to $p(\mathbf{x}_{\text{mis}}|\mathbf{x}_{\text{obs}})$ by comparing two groups of samples collectively.

As a concrete example, Figure 10 compares the two evaluation metrics on MNIST, our distribution-based FID and the ground truth-based RMSE. It shows that the rankings on most of the missing rates are not consistent across the two metrics. In particular, under 90% missing rate, MisGAN is worse than GAIN and matrix factorization in terms of RMSE, but significantly better in terms of FID. Figure 11 plots the imputation results of the three methods mentioned above. We can clearly see that MisGAN produces the best completion even though its RMSE is much higher than the other two. It's not surprising as the mean of $p(\mathbf{x}_{\text{mis}}|\mathbf{x}_{\text{obs}})$ minimizes the squared error in expectation, even if the mean might have low density. This probably explains why the blurry completion results produced by matrix factorization achieve the lowest RMSE.

## D    ARCHITECTURAL DETAILS AND HYPERPARAMETERS

All of the generators and discriminators in Conv-MisGAN follow the architecture used by the DCGAN model (Radford et al., 2015) with 128-dimensional latent code.

As for FC-MisGAN, the architecture of the generators is

$$\text{FC}(128, 256)\text{--FC}(256, 512)\text{--FC}(512, 784)$$

with ReLUs in between. The discriminators are of the structure

$$\text{FC}(784, 512)\text{--FC}(512, 256)\text{--FC}(256, 128)\text{--FC}(128, 1)$$

also with ReLUs in between.

For the imputer network for MisGAN trained on CIFAR-10 and CelebA, we follow the U-Net implementation of the CycleGAN and pix2pix work[6]. In the experiments, we use 5-layer U-Nets for both CIFAR-10 and CelebA.

For training Wasserstein GAN with gradient penalty, We use all the default hyperparameters reported in Gulrajani et al. (2017). For all the datasets, MisGAN is trained for 300 epochs. We train MisGAN imputer for 1000 epochs for MNIST and CIFAR-10 as the networks are smaller and 600 epochs for CelebA.

For ConvAC, we use the same architecture described in Sharir et al. (2016). We train ConvAC for 1000 epochs using Adam optimizer with learning rate $10^{-4}$.

## E    MISGAN ON CIFAR-10

Figure 12, 13 and 14 show the results of MisGAN trained on CIFAR-10 for the two extreme missing rates, namely 90% and 80%, as well as the case of 10% that is close to full observation.

## F    MISGAN ON CELEBA

Figure 15, 16 and 17 show the results of MisGAN trained on CelebA for the two extreme missing rates, namely 90% and 80%, as well as the case of 10% that is close to full observation.

## G    RESULTS OF CONVAC

Figure 18 shows the samples generated by ConvAC trained with the square observation missing pattern on MNIST.

## H    MISSING DATA IMPUTATION WITH GAIN

Figure 19 shows the imputation results of GAIN on different epochs during training with the $20 \times 20$ square observation missingnss. We found that this is a common phenomenon for the square observation missing pattern. To obtain better results for GAIN, we analyze the FIDs during the course of training and use the model that achieves the best FID to favorably compare with MisGAN for the square observation case. For CIFAR-10, we use the results from the 500th epoch; for CelebA, we use the results from the 50th epoch. Otherwise, we train GAIN for 1000 epochs for CIFAR-10 and 300 epochs for CelebA. Our implementation is adapted from the code released by the authors of GAIN.[7]

Figure 20 shows the imputation results of GAIN for both CIFAR-10 and CelebA.

---

[6] `https://github.com/junyanz/pytorch-CycleGAN-and-pix2pix`
[7] `https://github.com/jsyoon0823/GAIN`

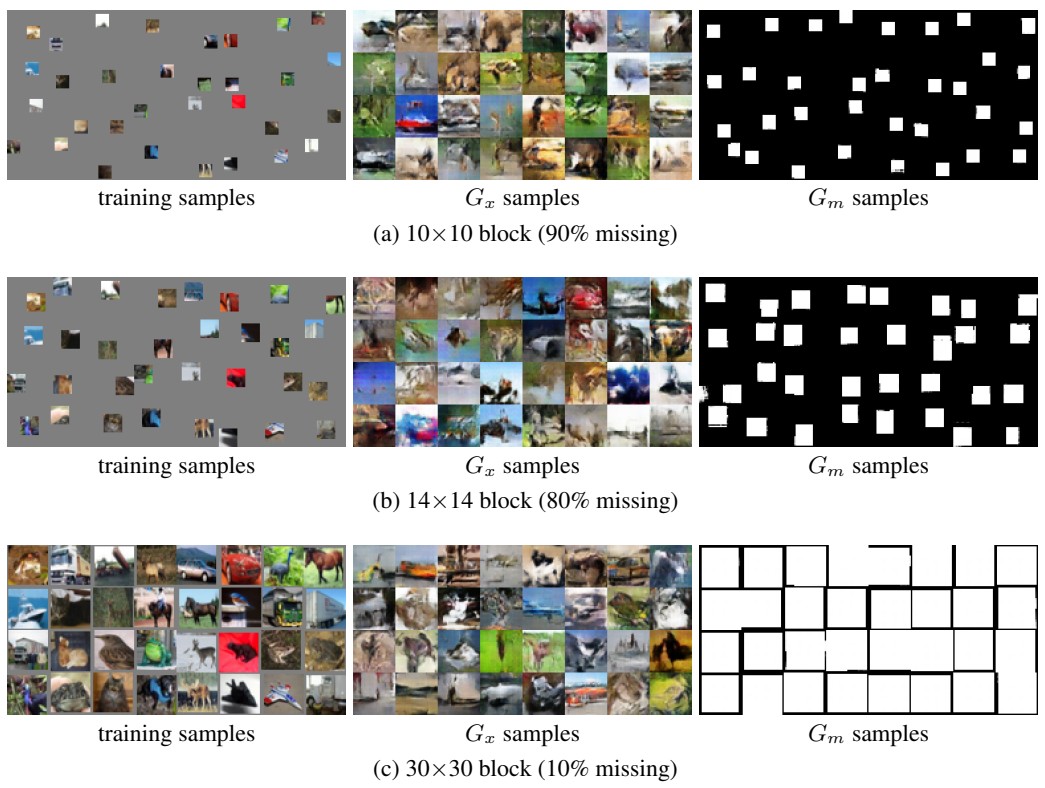

training samples        $G_x$ samples        $G_m$ samples

(a) 10×10 block (90% missing)

training samples        $G_x$ samples        $G_m$ samples

(b) 14×14 block (80% missing)

training samples        $G_x$ samples        $G_m$ samples

(c) 30×30 block (10% missing)

Figure 12: MisGAN on CIFAR-10 with block observation missingness

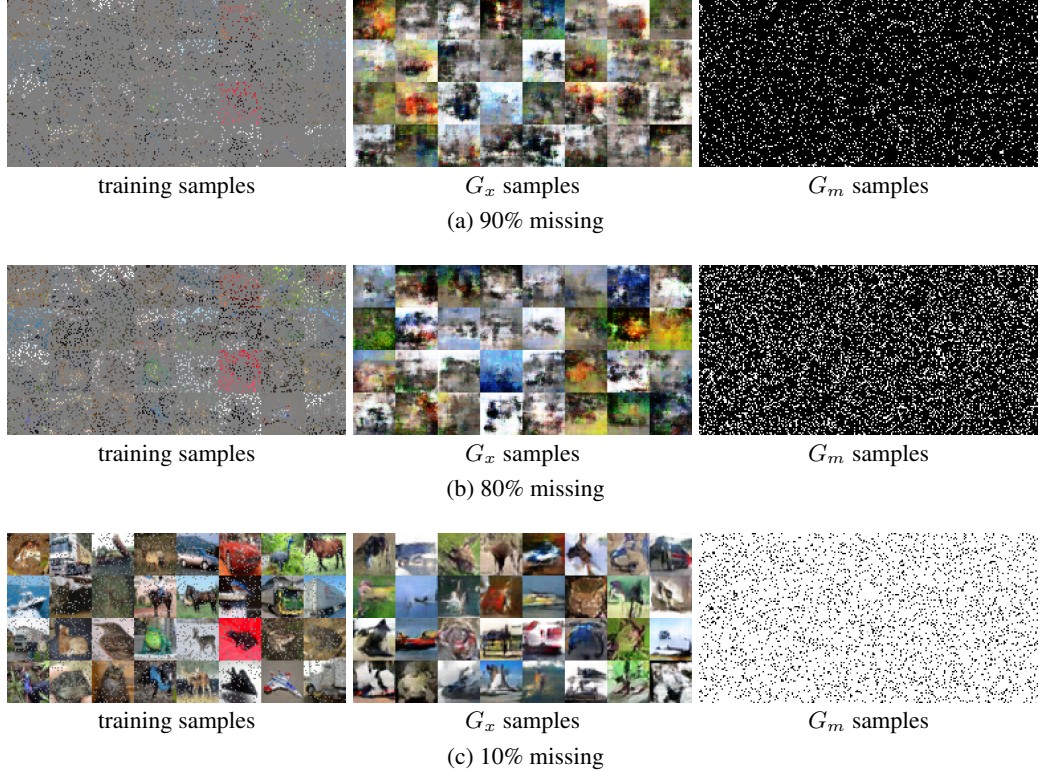

training samples        $G_x$ samples        $G_m$ samples

(a) 90% missing

training samples        $G_x$ samples        $G_m$ samples

(b) 80% missing

training samples        $G_x$ samples        $G_m$ samples

(c) 10% missing

Figure 13: MisGAN on CIFAR-10 with independent dropout missingness

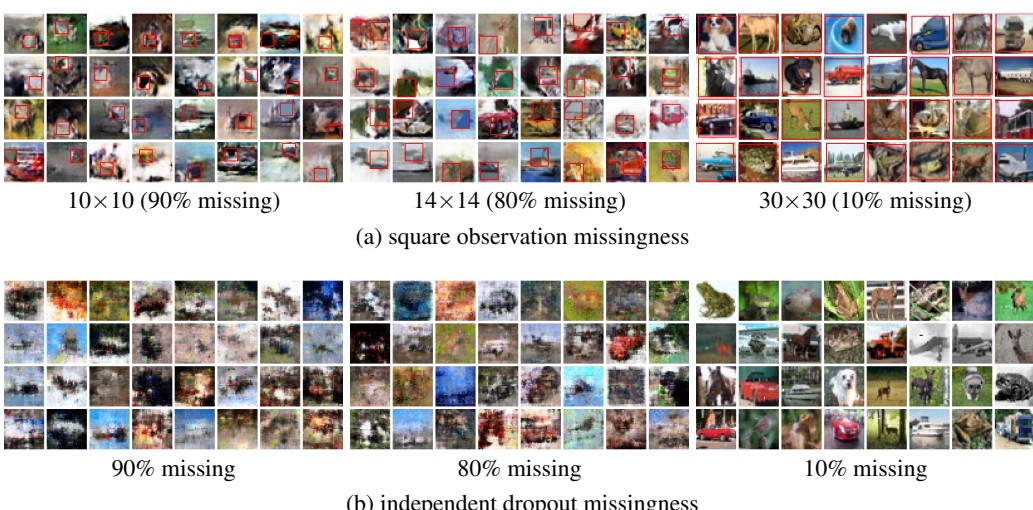

$10\times10$ (90% missing)    $14\times14$ (80% missing)    $30\times30$ (10% missing)

(a) square observation missingness

90% missing    80% missing    10% missing

(b) independent dropout missingness

Figure 14: MisGAN imputation on CIFAR-10

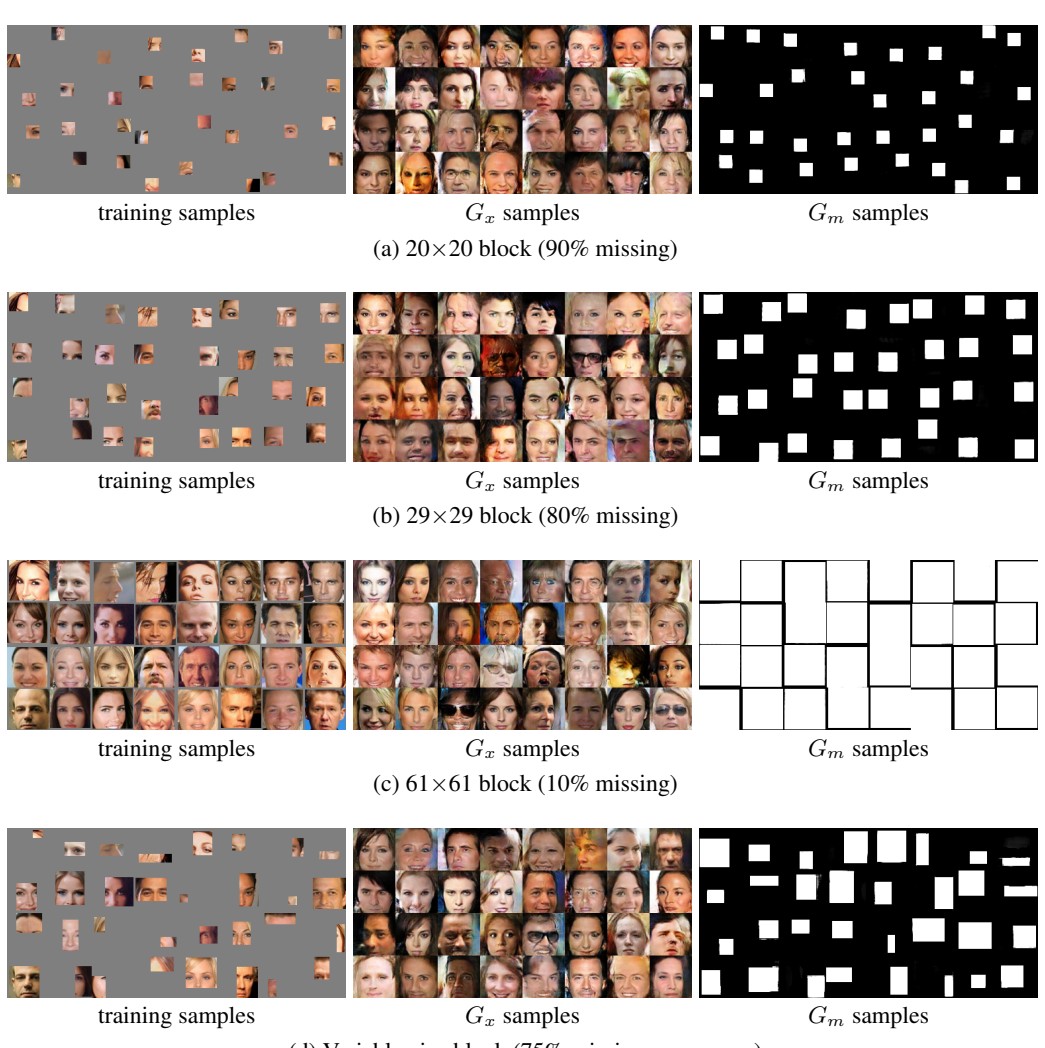

training samples    $G_x$ samples    $G_m$ samples

(a) $20\times20$ block (90% missing)

training samples    $G_x$ samples    $G_m$ samples

(b) $29\times29$ block (80% missing)

training samples    $G_x$ samples    $G_m$ samples

(c) $61\times61$ block (10% missing)

training samples    $G_x$ samples    $G_m$ samples

(d) Variable-size block (75% missing on average)

Figure 15: MisGAN on CelebA with block observation missingness

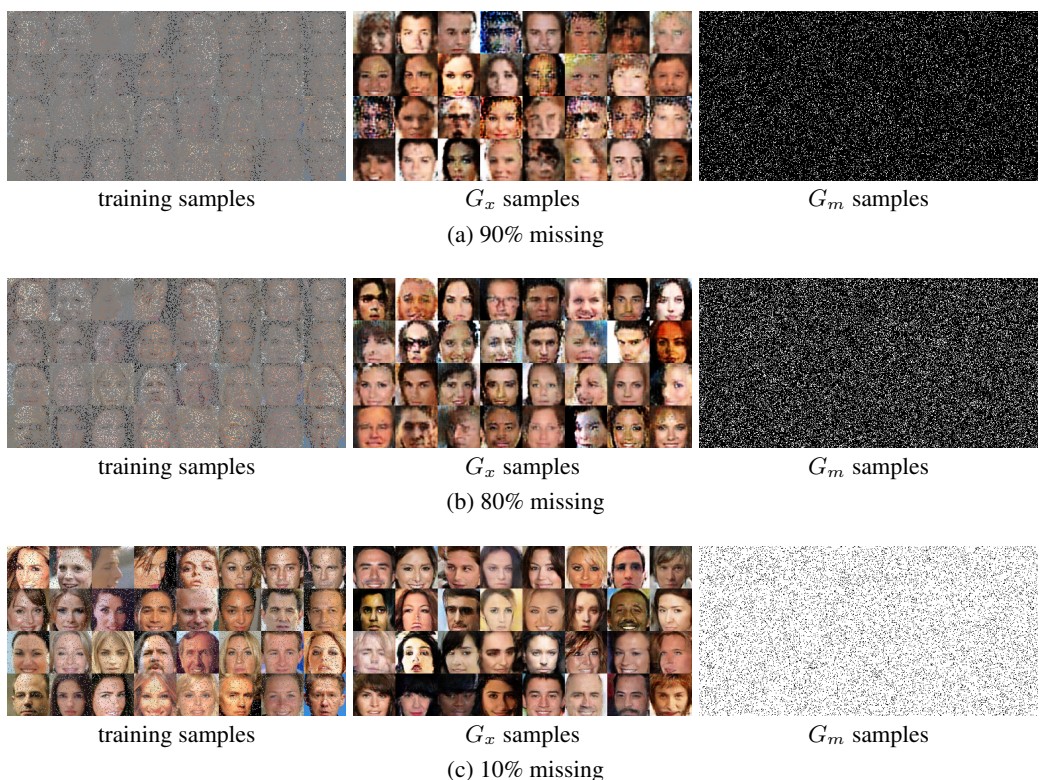

training samples      $G_x$ samples      $G_m$ samples

(a) 90% missing

training samples      $G_x$ samples      $G_m$ samples

(b) 80% missing

training samples      $G_x$ samples      $G_m$ samples

(c) 10% missing

Figure 16: MisGAN on CelebA with independent dropout missingness

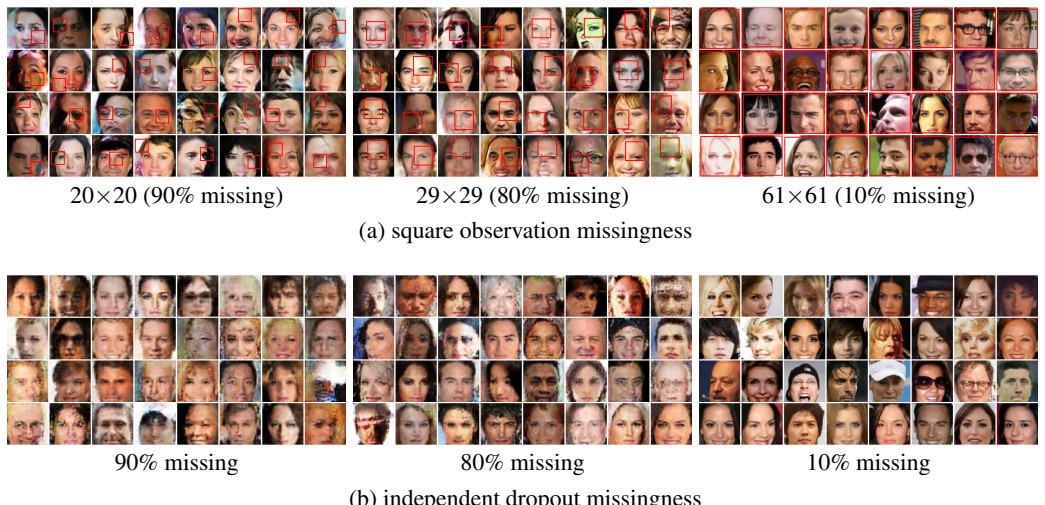

20×20 (90% missing)      29×29 (80% missing)      61×61 (10% missing)

(a) square observation missingness

90% missing      80% missing      10% missing

(b) independent dropout missingness

Figure 17: MisGAN imputation on CelebA

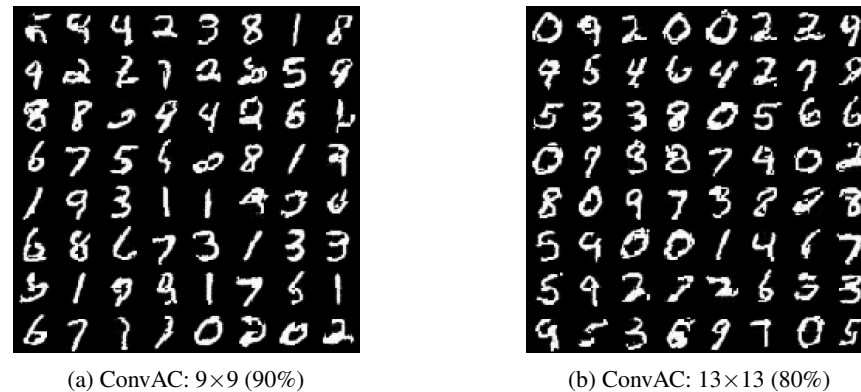

(a) ConvAC: 9×9 (90%)           (b) ConvAC: 13×13 (80%)

Figure 18: Results of ConvAC trained with square observations of different sizes on MNIST.

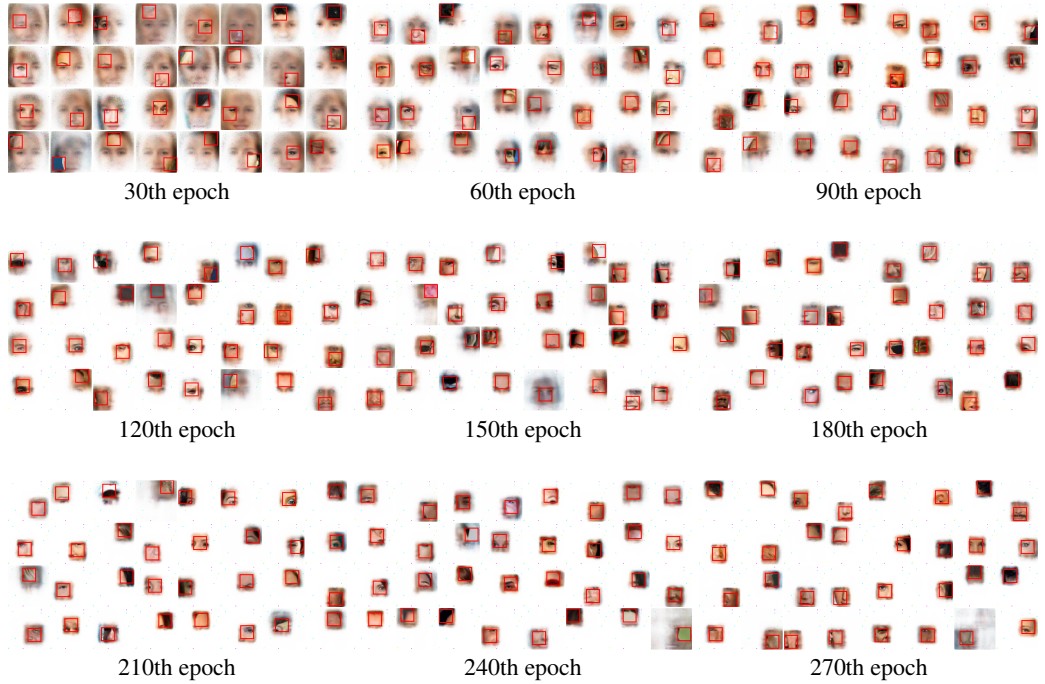

Figure 19: Imputation results of GAIN on different epochs during training under 20×20 square observation missingness. If over-trained, the imputation behavior of GAIN will gradually become similar to constant imputation.

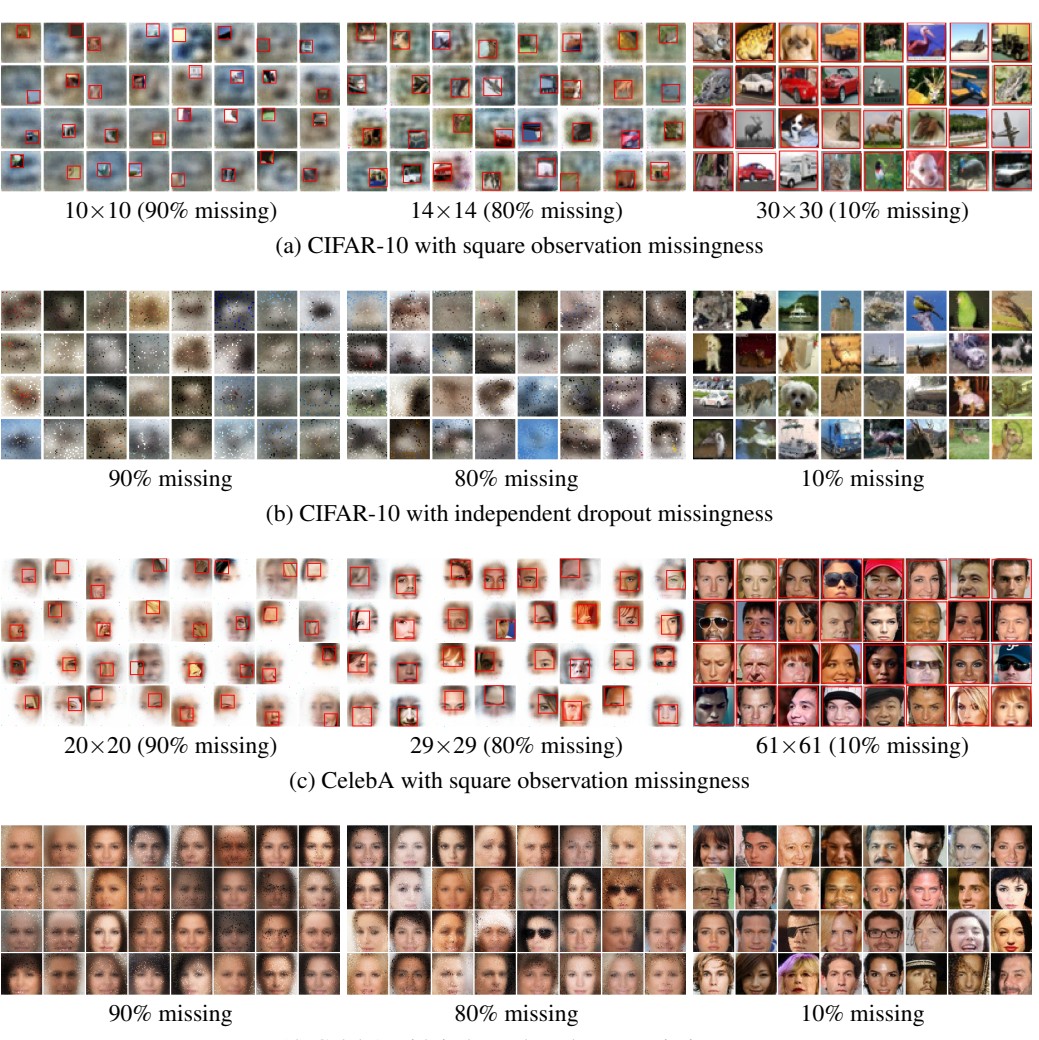

(a) CIFAR-10 with square observation missingness

(b) CIFAR-10 with independent dropout missingness

(c) CelebA with square observation missingness

(d) CelebA with independent dropout missingness

Figure 20: GAIN imputation

