# OpenReview forum: "MisGAN: Learning from Incomplete Data with Generative Adversarial Networks"
_ICLR.cc/2019/Conference_

### Official Review · AnonReviewer3 · 2018-11-03
**Nice extension of AmbientGAN with detail experiment analysis**

**Rating:** 7
**Confidence:** 4

**Review:**

This paper proposed a new network structure to learn GAN with incomplete data, and it is a nice extension of AmbientGAN. Two theorems are provided for better understanding the potential effect of the missing values. Improved results compared with state-of-the-art methods on MNIST, CIFAR-10 and CelebA are presented. Overall, the paper is well organized, and the experiment results are sufficient to demonstrate the advantages of the proposed method. I particular like figure5 where AmbientGAN failed in this case.

 Several suggestions about improving the paper. I notice the images used in the experiments are small size. It would be interesting to test the performance on a larger image. Another direction would be testing the robustness of the model, for example, what will happen if the observation is also noisy? Some discussion about the potential extensions will also be helpful. For example, can the proposed network be used to solve the compressive sensing problem with a real value mask instead of binary valued.

I did not dive into the detail of the prove of theorems. And it seems valid by reading through each step.  Although these two theorems are not directly related to the properties of the proposed network structure. But it does provide some nice intuition.

---

> ### Author Response · Authors · 2018-11-22
> **MisGAN is a flexible and extensible framework**
>
> Thank you for the constructive comments, which we address below.
>
> > I notice the images used in the experiments are small size. It would be interesting to test the performance on a larger image.
>
> Learning the distribution of high-resolution images poses a different challenge to GAN-based models that is orthogonal to missing data. However, the MisGAN framework is flexible enough to incorporate various techniques that can improve training GANs with large images such as Karras et al. (2017) or Brock et al. (2018).
>
> > Another direction would be testing the robustness of the model, for example, what will happen if the observation is also noisy?
>
> If the training data is noisy, the unmodified MisGAN would learn the distribution of noisy data as well. If we are interested in recovering the distribution of the denoised data, we can replace the data generator in MisGAN by an AmbientGAN with certain assumption on the noise structure. The recovery would depend on how accurate the noise assumption is. Note that if we use a complex noise model, the problem might become highly ill-posed and the learned distribution of the clean data can be quite different from the actual one as we showed in Figure 5. In this case, we will need to introduce some domain-specific priors to further regularize the problem.
>
> > Some discussion about the potential extensions will also be helpful. For example, can the proposed network be used to solve the compressive sensing problem with a real value mask instead of binary valued.
>
> The noise structure of the missing data has a special property that makes MisGAN possible: the missingness is fully observed. This provides a strong signal to regularize the originally highly ill-posed distribution learning problem. MisGAN might be applicable to other noise models that have similar properties.
>
> > Although these two theorems are not directly related to the properties of the proposed network structure. But it does provide some nice intuition.
>
> We emphasize that, as we pointed out at the end of Appendix A, an implication of the theorem is that MisGAN overall learns the joint distribution p(x_obs, m). This is different from most of the work in the literature that aims at modeling p(x_obs | m) instead, which ignores the missing data mechanism.
>
>
> REFERENCES
>
> Karras, T., Aila, T., Laine, S., & Lehtinen, J. (2017). Progressive growing of GANs for improved quality, stability, and variation.
> Brock, A., Donahue, J., & Simonyan, K. (2018). Large scale GAN training for high fidelity natural image synthesis.

---

### Official Review · AnonReviewer2 · 2018-11-05
**Resolving a major challenge in AmbientGAN, by focusing on a very specific application.**

**Rating:** 6
**Confidence:** 5

**Review:**

Building upon the success of AmbientGAN by Bora, Price, and Dimakis, this paper studies one of the major issues that is not resolved in AmbientGAN: the distribution of the data corruption is typically unknown. In general this is an ill-defined problem to solve, as the data corruption distribution is not identifiable from the corrupted data. The major insight of this paper is to identify a plausible setting where such identifiability issues are not present. Namely, the corruption itself is identifiable from the corrupted data. The brilliance of this paper is in identifying this niche application of data imputation/missing data/incomplete data.

Once the goal is set to train a GAN on incomplete data, the solution somewhat follows in a straightforward manner from AmbientGAN. Pass the generated output through a masking operator, which is also trained. Train the masking operator on the masking pattern of the real (corrupted) data. Imputation generator and discriminator also follows in a straightforward manner.

A major shortcoming of this paper is that the performance of the proposed approach is not fully supported by extensive experiments. For example, a major application of such imputation solution will be predicting missing data in real world applications, such as recommendation systems, or biological experimental data. A experimental setting in "GAIN: Missing Data Imputation using Generative Adversarial Nets" provides an excellent benchmark dataset, and imputation approaches should be compared against GAIN in those scenarios.

---

> ### Author Response · Authors · 2018-11-22
> **On the benchmark data used by GAIN**
>
> Thank you for the constructive comments, which we address below.
>
> As stated in the introduction, MisGAN is designed for learning the distribution from high-dimensional data in the presence of a potentially large amount of missing values. However, the five benchmark datasets that GAIN (Yoon et al, 2018) is evaluated on are quite small (see Table 1 in the supplementary materials of Yoon et al. (2018) for details). The average number of examples in a dataset is less than 20,000 and the smallest one has only 569 examples. Moreover, the dimensionality of the data is also relatively small, where the average dimensionality is about 36 while the smallest one is 16. As a result, GAN-based models like MisGAN are not particularly suitable for this situation. The table below compares the imputation RMSE of MisGAN with GAIN and MICE, one of the strong baseline GAIN compares to. The first two rows in the table directly come from Table 2 in Yoon et al. (2018), where the results of MICE (R) are computed using the R package MICE. We find that a popular Python implementation of MICE, fancyimpute, outperforms all the methods on all five datasets as shown in the third row (and it runs faster than the R implementation). However, MisGAN performs worse than GAIN on most of the cases due to the fact that the datasets are too small to learn a good data generator that drives imputation. On the other hand, data-efficient methods like MICE appear to be a better modeling choice when data is scarce. Nevertheless, learning distributions from small-scale incomplete data with MisGAN is an interesting direction for future investigation.
>
>                                        Breast    Spam    Letter    Credit    News
> GAIN                               .0546     .0513     .1198     .1858     .1441
> MICE (R)                         .0646     .0699     .1537     .2585     .1763
> MICE (fancyimpute)     .0498     .0494     .1126     .1217     .1426
> MisGAN                          .0855     .0637     .1632     .1656     .2442
>
> To better understand the imputation behavior on high-dimensional incomplete data, for which MisGAN targets, we choose to perform a set of controlled experiments with different missing distributions under a wide range of missing rates (unlike Yoon et al. (2018) that only assessed 20% missingness). We choose to evaluate on image datasets so we can visually judge if the evaluation agrees with our intuition as in Figure 21. Note that for the MNIST results in Figure 7, we use fully-connected imputers in MisGAN as if the model has no prior knowledge about the structure of the underlying data to demonstrate that MisGAN can be applied to generic data other than images.
>
> We note that fancyimpute’s MICE not only outperforms both MisGAN and GAIN on the benchmark datasets in Yoon et al. (2018) but runs much faster. However, it can hardly scale up to data like MNIST, although being only 784-dimensional.
>
>
> REFERENCES
>
> fancyimpute: https://github.com/iskandr/fancyimpute

---

### Official Review · AnonReviewer1 · 2018-11-08
**Good paper but need to rectify few things**

**Rating:** 7
**Confidence:** 4

**Review:**

This is a good paper, as we have good experimental evidence that the proposed method seems to have some advantage over baseline methods.

The authors measure the success of their algorithm by computing FID scores for the randomly inputed images. That is the authors use a metric which measures a distance between the distribution of the generated images and images in a dataset. This is fine and interesting to know, but people also care about the distance of the completed pixels from the ground truth (missing) values. (E.g. https://www.cs.rochester.edu/u/jliu/paper/Ji-ICCV09.pdf)

This is important, because in a real life application, one would pick the mode of the distribution of the missing samples, and not sample from that distribution as the authors seems to be doing in this paper.

I would therefore suggest adding experiments where authors pick the mode of the distribution and estimate an error metric such as root mean square error (RMSE or PSNR https://en.wikipedia.org/wiki/Peak_signal-to-noise_ratio )

I also found the 'marketing'/presentation of the algorithm little misleading, especially in the introduction, given that there exists another GAN based imputation algorithm. I think the authors should clearly state in the introduction that the other algorithm, abbreviated GAIN, exists as a GAN based missing data completion method. Then they should point out the differences of this algorithm from GAIN. Namely they should elaborate verbally on why learning the missing data distribution helps. Overall, what I am trying to say is, the key idea of this paper - that is learning the mask distribution - is not well motivated in this paper.

Despite my concerns above, I recommend an accept. The algorithm seems novel, and there is some experimental results to back it up.

---

> ### Author Response · Authors · 2018-11-22
> **Evaluation of imputation using RMSE**
>
> Thank you for the constructive comments, which we address below.
>
> > In a real life application, one would pick the mode of the distribution of the missing samples, and not sample from that distribution as the authors seems to be doing in this paper.
>
> In this work, the imputation model we proposed learns an implicit model that generates samples from p(x_mis | x_obs) where the density is not explicitly defined. This implies that the imputer is likely to generate samples around the modes of the distribution, although the modes are not explicitly known. However, MisGAN is compatible with density-based imputation methods as well. For example, we can define a density model similar to variational autoencoders (Kingma and Welling, 2014) that imposes an isotropic Gaussian noise at the end of the deterministic decoder. With such density model, we can then optimize the latent code using gradient methods to output the mode (local maxima of the density function) of p(x_mis | x_obs) or amortize density maximization using a similar imputation network. We leave the comprehensive evaluation of this alternative imputation method for future work.
>
> > The authors measure the success of their algorithm by computing FID scores for the randomly inputed images. That is the authors use a metric which measures a distance between the distribution of the generated images and images in a dataset. This is fine and interesting to know, but people also care about the distance of the completed pixels from the ground truth (missing) values.
>
> To evaluate the imputation performance in a controlled experiment, we chose to assess the FID between the imputed data and the originally fully-observed data instead of following the commonly used approach that computes the RMSE against the ground truth for the following reasons: In a complex system, the conditional distribution p(x_mis | x_obs) is likely to be highly multimodal. It’s not guaranteed that the ground truth of the missing features in the incomplete dataset created under the missing completely at random (MCAR) assumption correspond to the global mode of p(x_mis | x_obs). A good imputation model might generate samples associated with a higher density than the ground truth (or from other modes that are similarly probable). In this case, it will lead to a large error in terms of metrics like RMSE as multiple modes might be far away from each other in a complex distribution. On the other hand, our evaluation methods using FID provides a practical way to assess how close a model imputes according to p(x_mis | x_obs) by comparing distributions collectively.
>
> As a concrete example, Figure 20 in Appendix J (updated) compares the two evaluation metrics on MNIST: our distribution-based FID and the ground truth-based RMSE. It shows that the rankings under most of the missing rates we assessed are not consistent across the two metrics. In particular, under 90% missing rate, MisGAN is worse than GAIN and matrix factorization in terms of RMSE, but significantly better in terms of FID. Figure 21 plots the imputation results of the three methods mentioned above. We can see that MisGAN produces the most visually promising completion even though its RMSE is much higher than the other two. It’s not surprising as the mean of p(x_mis | x_obs) minimizes the squared error in expectation, even if it might have low density. This probably explains why the blurry completion results produced by matrix factorization achieve the lowest RMSE.
>
> > I found the 'marketing'/presentation of the algorithm little misleading, especially in the introduction, given that there exists another GAN based imputation algorithm.
>
> Unlike GAIN, the main goal of this work, as we stated in the introduction, is trying to learn the distribution from high-dimensional incomplete data, which is applicable to a broader range of tasks other than missing data imputation. For example, we can train the model with interpretable priors in the generator network for better exploratory analysis. Moreover, a good generative model usually provides simpler or more effective algorithms for missing data imputation. For example, we can instead follow the imputation procedure in Rezende et al. (2014) accompanied with MisGAN to handle the situation when the missing data mechanism is different from the training distribution.
>
>
> REFERENCES
>
> Kingma, D. P., & Welling, M. (2014). Auto-encoding variational bayes.
> Rezende, D. J., Mohamed, S., & Wierstra, D. (2014). Stochastic backpropagation and approximate inference in deep generative models.

---

### Meta-Review · Area_Chair1 · 2018-12-14
**Intersting idea with practical impact**

**Confidence:** 3
**Recommendation:** Accept (Poster)

**Metareview:**

The paper proposes an adversarial framework that learns a generative model along with a mask generator to model missing data and by this enables a GAN to learn from incomplete data.
The method builds on AmbientGAN but it is a novel and clever adjustment to the specific problem setting of learning from incomplete data, that is of high practical interest.